# Neighborhood-aware Scalable Temporal Network Representation Learning

**Yuhong Luo**[*]
University of Massachusetts
yuhongluo@umass.edu

**Pan Li**
Purdue University
panli@purdue.edu

## Abstract

Temporal networks have been widely used to model real-world complex systems such as financial systems and e-commerce systems. In a temporal network, the joint neighborhood of a set of nodes often provides crucial structural information useful for predicting whether they may interact at a certain time. However, recent representation learning methods for temporal networks often fail to extract such information or depend on online construction of structural features, which is time-consuming. To address the issue, this work proposes **N**eighborhood-**A**ware **T**emporal network model (NAT). For each node in the network, NAT abandons the commonly-used one-single-vector-based representation while adopting a novel *dictionary-type neighborhood representation*. Such a dictionary representation records a down-sampled set of the neighboring nodes as keys, and allows fast construction of structural features for a joint neighborhood of multiple nodes. We also design a dedicated data structure termed *N-cache* to support parallel access and update of those dictionary representations on GPUs. NAT gets evaluated over seven real-world large-scale temporal networks. NAT not only outperforms all cutting-edge baselines by averaged 1.2%↑ and 4.2%↑ in transductive and inductive link prediction accuracy, respectively, but also keeps scalable by achieving a speed-up of 4.1-76.7× against the baselines that adopt joint structural features and achieves a speed-up of 1.6-4.0× against the baselines that cannot adopt those features. The link to the code: `https://github.com/Graph-COM/Neighborhood-Aware-Temporal-Network`.

## 1 Introduction

Temporal networks are widely used as abstractions of real-world complex systems [1]. They model interacting elements as nodes, interactions as links, and when those interactions happen as timestamps on those links. Temporal networks often evolve by following certain patterns. Ranging from triadic closure [2] to higher-order motif closure [3–6], the interacting behaviors between multiple nodes have been shown to strongly depend on the network structure of their joint neighborhood. Researchers have leveraged this observation and built many practical systems to monitor and make prediction on temporal networks such as anomaly detection in financial networks [7–9], friend recommendation in social networks [10], and collaborative filtering techniques in e-commerce systems [11].

Recently, graph neural networks (GNNs) have been widely used to encode network-structured data [12] and have achieved state-of-the-art (SOTA) performance in many tasks such as node/graph classification [13–15]. However, to predict how nodes interact with each other in temporal networks, a direct generalization of GNNs may not work well. Traditional GNNs often learn a vector representation for each node, and predict whether two nodes may interact (aka. a link) based on a combination (e.g. the inner product) of the two vector representations. This link prediction strategy often fails to capture the structural features of the joint neighborhood of the two nodes [16–19]. Consider a toy example with a temporal network in Fig. 1: Node $w$ and node $v$ share the same local structure before $t_3$, so GNNs including their variants on temporal networks (e.g., TGN [20]) will associate $w$

---

[*]This project is completed during Yuhong Luo's summer internship at Purdue University.

Y. Luo et al., Neighborhood-aware Scalable Temporal Network Representation Learning. *Proceedings of the First Learning on Graphs Conference (LoG 2022)*, PMLR 198, Virtual Event, December 9–12, 2022.

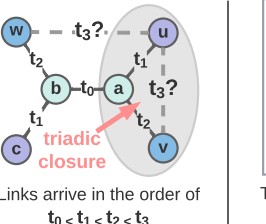 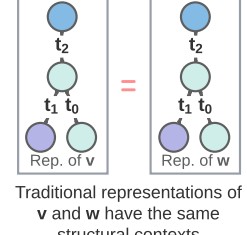 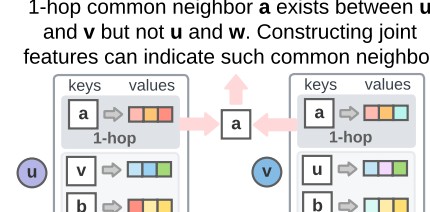

**Figure 1:** A toy example to predict how a temporal network evolves. Given the historical temporal network as shown in the left, the task is to predict whether $u$ prefers to interact with $v$ or $w$ at timestamp $t_3$. If this is a social network, $(u, v)$ is likely to happen because $u, v$ share a common neighbor $a$ and follow the principle of triadic closure [2]. However, traditional GNNs, even for their generalization on temporal networks fail here as they learn the same representations for node $v$ and node $w$ due to their common structural contexts, as shown in the middle. In the right, we show a high-level abstraction of joint neighborhood features based on N-caches of **u** and **v**: In the N-caches for 1-hop neighborhoods of both node $u$ and node $v$, $a$ appears as the keys. Joining these keys can provide a structural feature that encodes such common-neighbor information at least for prediction.

and $v$ with the same vector representation. Hence, GNNs will fail to make a correct prediction to tell whether $u$ will interact with $w$ or $v$ at $t_3$. Here, GNNs cannot capture the important joint structural feature that $u$ and $v$ have a common neighbor $a$ before $t_3$. This issue makes almost all previous works that generalize GNNs for temporal networks provide only subpar performance [20–29].

Some recent works have been proposed to address such an issue on static networks [18, 19, 30]. Their key ideas are to construct node structural features to learn the two-node joint neighborhood representations. Specifically, for two nodes of interest, they either label one linked node and construct its distance to the other node [31, 32], or label all nodes in the neighborhood with their distances to these two linked nodes [18, 33]. Traditional GNNs can afterward encode such feature-augmented neighborhood to achieve better inference. Although these ideas are theoretically powerful [18, 19] and provide good empirical performance on small networks, the induced models are not scaled up to large networks. This is because constructing such structural features is time-consuming and should be done separately for each link to be predicted. This issue becomes even more severe over temporal networks, because two nodes may interact many times and thus the number of links to be predicted is often much larger than the corresponding number in static networks.

In this work, we propose **N**eighborhood-**A**ware **T**emporal network model (NAT) that can address the aforementioned modeling issue while keeping good scalability of the model. The key novelty of NAT is to incorporate dictionary-type neighborhood representations in place of one-single-vector node representation and a computation-friendly neighborhood cache (N-cache) to maintain such dictionary-type representations. Specifically, the N-cache of a node stores several size-constrained dictionaries on GPUs. Each dictionary has a sampled collection of historical neighbors of the center node as keys, and aggregates the timestamps and the features on the links connected to these neighbors as values (vector representations). With N-caches, NAT can in parallel construct the joint neighborhood structural features for a batch of node pairs to achieve fast link predictions. NAT can also update the N-caches with new interacted neighbors efficiently by adopting hash-based search functions that support GPU parallel computation.

NAT provides a novel solution for scalable temporal network representation learning. We evaluate NAT over 7 real-world temporal networks, among which, one contains 1M+ nodes and almost 10M temporal links to evaluate the scalability of NAT. NAT outperforms cutting-edge baselines by averaged 1.2%↑ and 4.2%↑ in transductive and inductive link prediction accuracy respectively. NAT achieves 4.1-76.7× speed-up compared to the baseline CAWN [34] that constructs joint neighborhood features based on random walk sampling. NAT also achieves 1.6-4.0× speed-up of the fastest baselines that do not construct joint neighborhood features (and thus suffer from the issue in Fig. 1) on large networks.

## 2   Related works

Neighborhood structure often governs how temporal networks evolve over time. Early-time temporal network prediction models count motifs [35, 36] or subgraphs [37] in the historical neighborhood

of two interacting objects as features to predict their future interactions. These models cannot use network attributes and often suffer from scalability issues because counting combinatorial structures is complicated and hard to be executed in parallel. Network-embedding approaches for temporal networks [38–42] suffer from the similar problem, because the optimization problem used to compute node embeddings is often too complex to be solved again and again as the network evolves.

Recent works based on neural networks often provide more accurate and faster models, which benefit from the parallel computation hardware and scalable system support [43, 44] for deep learning. Some of these works simply aggregate the sequence of links into network snapshots and treat temporal networks as a sequence of static network snapshots [21–26]. These methods may offer low prediction accuracy as they cannot model the interactions that lie in different levels of time granularity.

Move advanced methods deal with link streams directly [20, 27–29, 45–48]. They generalize GNNs to encode temporal networks by associating each node with a vector representation and update it based on the nodes that one interacts with. Some works use the representation of the node that one is currently interacting with [27, 28, 45]. Other works use those of the nodes that one has interacted with in history [20, 29, 46, 47]. However, in either way, these methods suffer from the limited power of GNNs to capture the structural features from the joint neighborhood of multiple nodes [17, 19]. Recently, CAWN [34] and HIT [4], inspired by the theory in static networks [18, 19], have proposed to construct such structural features to improve the representation learning on temporal networks, CAWN for link prediction and HIT for higher-order interaction prediction. However, their computational complexity is high, as for every queried link, they need to sample a large group of random walks and construct the structural features on CPUs that limit the level of parallelism. However, NAT addresses these problems via neighborhood representations and N-caches.

# 3 Preliminaries: Notations and Problem Formulation

In this section, we introduce some notations and the problem formulation. We consider temporal network as a sequence of timestamped interactions between pairs of nodes.

**Definition 3.1 (Temporal network)** *A temporal network $\mathcal{E}$ can be represented as $\mathcal{E} = \{(u_1, v_1, t_1), (u_2, v_2, t_2), \cdots\}$, $t_1 \leq t_2 \leq \cdots$ where $u_i, v_i$ denote interacting node IDs of the $i$th link, $t_i$ denotes the timestamp. Each temporal link $(u, v, t)$ may have link feature $e_{u,v}^t$. We also denote the entire node set as $\mathcal{V}$. Without loss of generality, we use integers as node IDs, i.e., $\mathcal{V} = \{1, 2, ...\}$.*

A good representation learning of temporal networks is able to efficiently and accurately predict how temporal networks evolve over time. Hence, we formulate our problem as follows.

**Definition 3.2 (Problem formulation)** *Our problem is to learn a model that may use the historical information before $t$, i.e., $\{(u', v', t') \in \mathcal{E} | t' < t\}$, to accurately and efficiently predict whether there will be a temporal link between two nodes at time $t$, i.e., $(u, v, t)$.*

Next, we define *neighborhood* in temporal networks.

**Definition 3.3 ($k$-hop neighborhood in a temporal network)** *Given a timestamp $t$, denote a static network constructed by all the temporal links before $t$ as $\mathcal{G}_t$. Remove all timestamps in $\mathcal{G}_t$. Given a node $v$, define $k$-hop neighborhood of $v$ before time $t$, denoted by $\mathcal{N}_v^{t,k}$, as the set of all nodes $u$ such that there exists at least one walk of length $k$ from $u$ to $v$ over $\mathcal{G}_t$. For two nodes $u$, $v$, their joint neighborhood up-to $K$ hops refers to $\cup_{k=1}^K (\mathcal{N}_v^{t,k} \cup \mathcal{N}_u^{t,k})$.*

# 4 Methodology

In this section, we introduce NAT. NAT consists of two major components: neighborhood representations and N-caches, constructing joint neighborhood features and NN-based encoding.

## 4.1 Neighborhood Representations and N-caches

In NAT, a node representation is tracked by a fixed-sized memory module, i.e., N-cache over time as the temporal network evolves. Fig. 2 Left gives an illustration. In contrast to all previous methods that adopt a single vector representation for each node $u$, NAT adopts neighborhood representations $(Z_u^{(0)}(t), Z_u^{(1)}(t), ..., Z_u^{(K)}(t))$, where $Z_u^{(k)}(t)$ denotes the $k$-hop neighborhood representation, for

| No. | Notations | Definitions |
|---|---|---|
| 1. | $Z_u^{(k)}$ | A dictionary (with values $Z_{u,a}^{(k)}$ of size $M_k$) denoting the $k$-hop neighborhood representation for node $u$. |
| 2. | $Z_{u,a}^{(k)}$ | A vector (of length $F$ for $k \geq 1$) in the values of $Z_u^{(k)}$ representing node $a$ as a $k$-hop neighbor of $u$. |
| 3. | $s_u^{(k)}$ | An auxiliary array to record the node IDs who are currently recorded as the keys of $Z_u^{(k)}$. |
| 4. | $\mathrm{DE}_u^t(a)$ | The distance encoding of node $a$ based on the keys of N-caches of node $u$ at time $t$ (Eq. (1)). |
| 5. | $\mathrm{hash}(a)$ | The hash function mapping a node ID $a$ to the position of $Z_{u,a}^{(k)}$ in the $k$-hop N-cache of any node $u$. |

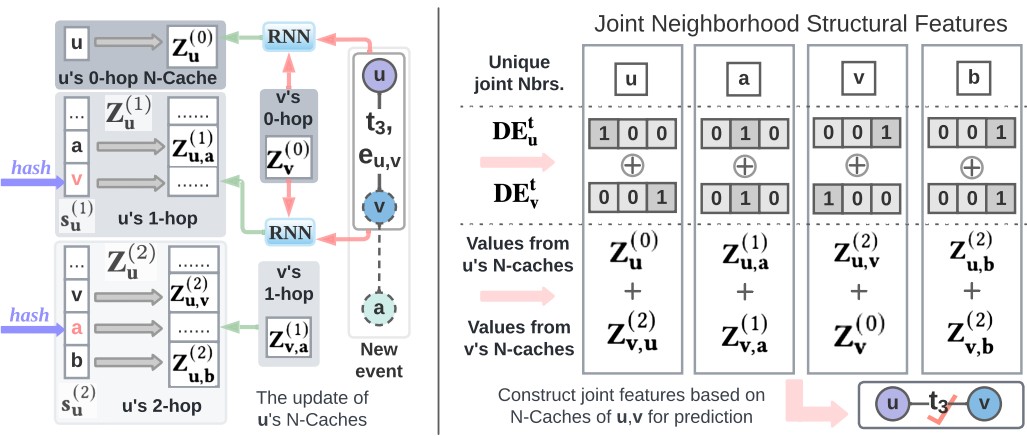

**Figure 2:** Neighborhood representations and Joining Neighborhood Features & Representations to make predictions. Left: Neighborhood representations of a node. Node $u$ interacts with $v$ at $t_3$ in the example in Fig. 1. The 0-hop (self) representation and 1-hop representations will be updated based on $Z_v^{(0)}$. The 2-hop representations will be updated by inserting $Z_v^{(1)}$. $Z_u^{(k)}$'s are maintained in N-caches. Right: In the example of Fig. 1, to predict the link $(u, v, t_3)$, the neighborhood representations of node $u$ and node $v$ will be joined: The structural feature DE is constructed according to Eq. (1); The representations are sum-pooled according to Eq. (2). Then, an attention layer (Eq. (3)) is adopted to make the final prediction. $\oplus$ denotes vector concatenation.

$k = 0, 1, ..., K$. Note that these representations may evolve over time. For notation simplicity, the timestamps in these notations are ignored while they typically can be inferred from the context. The main goal of tracking these neighborhood representations is to enable efficient construction of structural features, which will be detailed in Sec. 4.2. Next, we first explain these neighborhood representations from the perspective of modeling and how they evolve over time. Then, we introduce the scalable implementation of N-caches.

**Modeling.** For a node $u$, the 0-hop representation, or termed self-representation $Z_u^{(0)}$ simply works as the standard node representation for $u$. It gets updated via an RNN $Z_u^{(0)} \leftarrow \mathbf{RNN}(Z_u^{(0)}, [Z_v^{(0)}, t_3, e_{u,v}])$ when node $u$ interacts with another node $v$ as shown in Fig. 2 Left.

The rest neighborhood representations are more complicated. To give some intuition, we first introduce the 1-hop representation $Z_u^{(1)}$. $Z_u^{(1)}$ is a dictionary whose keys, denoted by key($Z_u^{(1)}$), correspond to a down-sampled set of the (IDs of) nodes in the 1-hop neighborhood of $u$. For a node $a$ in key($Z_u^{(1)}$), the dictionary value denoted by $Z_{u,a}^{(1)}$ is a vector representation as a summary of previous interactions between $u$ and $a$. $Z_u^{(1)}$ will be updated as temporal network evolves. For example, in Fig. 1, as $v$ interacts with $u$ at time $t_3$ with the link feature $e_{u,v}$, the entry in $Z_u^{(1)}$ that corresponds to $v$, $Z_{u,v}^{(1)}$ will get updated via an RNN $Z_{u,v}^{(1)} \leftarrow \mathbf{RNN}(Z_{u,v}^{(1)}, [Z_v^{(0)}, t_3, e_{u,v}])$. If $Z_{u,v}^{(1)}$ does not exist in current $Z_u^{(1)}$ (e.g., in the first $v, u$ interaction), a default initialization of $Z_{u,v}^{(1)}$ is used. Once updated, the new value $Z_{u,v}^{(1)}$ paired with the key (node ID) $v$ will be inserted into $Z_u^{(1)}$.

One remark is that for the input timestamps $t_i$, we adopt Fourier features to encode them before filling them into RNNs, i.e., with learnable parameter $\omega_i$'s, $1 \leq i \leq d$, T-encoding$(t) = [\cos(\omega_1 t), \sin(\omega_1 t), ..., \cos(\omega_d t), \sin(\omega_d t)]$, which has been proved to be useful for temporal network representation learning [4, 20, 29, 34, 49, 50].

The large-hop ($>1$) neighborhood representation $Z_u^{(k)}$ is also a dictionary. Similarly, the keys of $Z_u^{(k)}$ correspond to the nodes who lie in the $k$-hop neighborhood of $u$. The update of $Z_u^{(k)}$ is as

---

**Algorithm 1:** N-caches construction and update $(\mathcal{V}, \mathcal{E}, \alpha)$

---

1 **for** *k from 0 to 2 (consider only two hops)* **do**
2     **for** *u in $\mathcal{V}$,* **in parallel,** **do**
3        Initialize fixed-size dictionaries $Z_u^{(k)}$ in GPU with key spaces $s_u^{(k)}$ and value spaces;

4 **for** $(u, v, t, e)$ *in each mini-batch* $(\mathbf{u}, \mathbf{v}, \mathbf{t}, \mathbf{e})$ *of $\mathcal{E}$,* **in parallel,** **do**
5     $Z_u^{(0)} \leftarrow \mathbf{RNN}(Z_u^{(0)}, [Z_v^{(0)}, t, e])$ // update 0-hop self-representation
6     $Z_{\mathbf{prev}} \leftarrow Z_{u,v}^{(1)}$ **if** $s_u^{(1)}[hash(v)]$ **equals** $v$, **else 0** // check if $Z_{u,v}^{(1)}$ is recorded in $Z_u^{(1)}$ or not;
7     **if** $s_u^{(1)}[hash(v)]$ *equals ($v$ or EMPTY) or rand*$(0, 1) < \alpha$ **then**
8        $s_u^{(1)}[hash(v)] \leftarrow v$, $Z_{u,v}^{(1)} \leftarrow \mathbf{RNN}(Z_{\mathbf{prev}}, [Z_v^{(0)}, t, e])$; // update 1-hop nbr. representation
9     **for** $w$ *in* $s_v^{(1)}$*,* **in parallel,** **do**
10        **if** $s_u^{(2)}[hash(w)]$ *equals ($w$ or EMPTY) or rand*$(0, 1) < \alpha$ **then**
11           $s_u^{(2)}[hash(w)] \leftarrow w$, $Z_{u,w}^{(2)} \leftarrow Z_{v,w}^{(1)}$; // update 2-hop nbr. representations
12     **repeat lines 5-11** with $(v, u, t, e)$

---

follows: If $u$ interacts with $v$, $v$'s $(k-1)$-hop neighborhood by definition becomes a part of $k$-hop neighborhood of $u$ after the interaction. Given this observation, $Z_u^{(k)}$ can also be updated by using $Z_v^{(k-1)}$. However, we avoid using an RNN for the large-hop update to reduce complexity. Instead, we directly insert $Z_v^{(k-1)}$ into $Z_u^{(k)}$, i.e., setting $Z_{u,a}^{(k)} \leftarrow Z_{v,a}^{(k-1)}$ for all $a \in \text{key}[Z_v^{(k-1)}]$. If $Z_{u,a}^{(k)}$ has already existed before the insertion, we simply replace it.

Next, we will introduce the implementation of the above representations via N-caches. Readers who only care about the learning models can skip this part and directly go to Sec. 4.2. The maintenance of N-caches (aka. neighborhood representations) as the network evolves is summarized in Alg. 1.

**Scalable Implementation.** Neighborhood representations cannot be directly implemented via built-in hash tables such as *python dictionary* to achieve scalable maintenance. To maximize memory efficiency and parallelism, we adopt the following three design techniques: (a) Setting size limit; (b) Parallelizing hash-maps; (c) Addressing collisions.

*(a) Limiting size:* In a real-world network, the size of the neighborhood of a node typically follows a long-tailed distribution [51, 52]. So, it is irregular and memory inefficient to record the entire neighborhood. Instead, we set an upper limit $M_k$ to the size of each-hop representation $Z_u^{(k)}$, which means $Z_u^{(k)}$ may record only a subset of nodes in the $k$-hop neighborhood of node $u$. This idea is inspired by previous works that have shown structural features constructed based on a down-sampled neighborhood is sufficient to provide good performance [34, 53]. To further decrease the memory overhead, we only set each representation $Z_{u,a}^{(k)}, k \geq 1$ as a vector of small dimension $F$. Overall, the memory overhead of the N-cache per node is $O(\sum_{k=1}^{K} M_k \times F)$. In our experiments, we consider at most $K = 2$ hops, and set the numbers of tracked neighbors $M_1, M_2 \in [2, 40]$ and the size of each representation $F \in [2, 8]$, which already gives a very good performance. Based on the above design, the overall memory overhead is just about hundreds per node, which is comparable to the commonly-used memory cost of tracking a big single-vector representation for each node.

*(b) The hash-map:* As NAT needs to frequently access N-caches, a fast implementation of using node IDs to search within N-caches in parallel is needed. To enable the parallel search, we design GPU dictionaries to implement N-caches. Specifically, for every node $u$, we pre-allocate $O(M_k \times F)$ space in GPU-RAM to record the values in $Z_u^{(k)}$. A hash function is adopted to access the values in $Z_u^{(k)}$. For some node $a$, we compute $\text{hash}(a) \equiv (q * a) \ (\textbf{mod} \ M_k)$ for a fixed large prime number $q$ to decide the row-index in $Z_u^{(k)}$ that records $Z_{u,a}^{(k)}$. Such a simple hashing allows NAT accessing multiple neighborhood representations in N-caches in parallel.

However, as the size $M_k$ of each N-cache is small, in particular smaller than the corresponding neighborhood, the hash-map may encounter collisions. To detect such collisions, we also pre-allocate $O(M_k)$ space in each N-cache $Z_u^{(k)}$ for an array $s_u^{(k)}$ to record the IDs of the nodes who are the most

recent ones recorded in $Z_u^{(k)}$. Specifically, we use $s_u^{(k)}[\text{hash}(a)]$ to check whether node $a$ is a key of $Z_u^{(k)}$. If $s_u^{(k)}[\text{hash}(a)]$ is $a$, $Z_{u,a}^{(k)}$ is recorded at the position $\text{hash}(a)$ of $Z_u^{(k)}$. If $s_u^{(k)}[\text{hash}(a)]$ is neither $a$ nor EMPTY, the position $\text{hash}(a)$ of $Z_u^{(k)}$ records the representation of another node.

*(c) Addressing collisions:* If encountering a collision when NAT works on an evolving network, NAT addresses that collision efficiently with replacement in a random manner. Specifically, suppose we are to write $Z_{u,a}^{(k)}$ into $Z_u^{(k)}$. If another node $b$ satisfies $\text{hash}(a) = \text{hash}(b) = p$ and $Z_{u,b}^{(k)}$ has occupied the position $p$ of $Z_u^{(k)}$, then, we replace $Z_{u,b}^{(k)}$ by $Z_{u,a}^{(k)}$ (and $s_u^{(k)}[\text{hash}(a)] \leftarrow a$ simultaneously) with probability $\alpha$. Here, $\alpha \in (0, 1]$ is a hyperparameter. Although the above random replacement strategy sounds heuristic, it is essentially equivalent to random-sampling nodes from the neighborhood without replacement (random dropping $\leftrightarrow$ random sampling). Note that random-sampling neighbors is an effective strategy used to scale up GNNs for static networks [54–56], so here we essentially apply an idea of similar spirit to temporal networks. We find a small size $M_k$ ($\leq 40$) can give a good empirical performance while keeping the model scalable, and NAT is relatively robust to a wide range of $\alpha$.

## 4.2  Joint Neighborhood Structural Features and Neural-network-based Encoding

As illustrated in the toy example in Fig. 1, structural features from the joint neighborhood are critical to reveal how temporal networks evolve. Previous methods in static networks adopt distance encoding (DE) (or called labeling tricks more broadly) to formulate these features [18, 19]. Recently, this idea has got generalized to temporal networks [34]. However, the model CAWN in [34] uses online random-walk sampling, which cannot be parallelized on GPUs and is thus extremely slow. Our design of N-caches allows for addressing such a problem. Fig. 2 Right illustrates the procedure.

NAT generates joint neighborhood structural features as follows. Suppose our prediction is made for a temporal link $(u, v, t)$. For every node $a$ in the joint neighborhood of $u$ and $v$ decided by their N-caches at timestamp $t$, i.e., $a \in \left[ \cup_{k=0}^{K} \text{key}(Z_u^{(k)}) \right] \cup \left[ \cup_{k'=0}^{K} \text{key}(Z_v^{(k')}) \right]$, we associate it with a DE

$$\text{DE}_{uv}^t(a) = \text{DE}_u^t(a) \oplus \text{DE}_v^t(a), \text{ where } \text{DE}_w^t(a) = \left[ \chi[a \in Z_w^{(0)}], ..., \chi[a \in Z_w^{(K)}] \right], w \in \{u, v\}. \quad (1)$$

Here, $\chi[a \in Z_w^{(i)}]$ is 1 if $a$ is among the keys of N-cache $Z_w^{(i)}$ or 0 otherwise. $\oplus$ denotes vector concatenation. As for the example to predict $(u, v, t_3)$ in Fig. 1, the DEs of four nodes $u, a, v, b$ are as shown in Fig. 2 Right. Note that $\text{DE}_{uv}^{t_3}(a) = [0, 1, 0] \oplus [0, 1, 0]$ because $a$ appears in the keys of both $Z_u^{(1)}$ and $Z_v^{(1)}$, which further implies $a$ as a common neighbor of $u$ and $v$.

Simultaneously, NAT also aggregates neighborhood representations for every node $a$ in the common neighborhood of $u$ and $v$. Specifically, for node $a$, we aggregate the representations via a sum pool

$$Q_{uv}^t(a) = \sum_{k=0}^{K} \sum_{w \in \{u, v\}} Z_{w,a}^{(k)} \times \chi[a \in Z_w^{(k)}]. \quad (2)$$

Here, if $a$ is not in the neighborhood $Z_w^{(k)}$, $\chi[a \in Z_w^{(k)}] = 0$ and thus $Z_{w,a}^{(k)}$ does not participate in the aggregation. Both DE (Eq (1)) and representation aggregation (Eq (2)) can be done for multiple node pairs in parallel on GPUs. We detail the parallel steps in Appendix A. After joining DE and neighborhood representations, for each link $(u, v, t)$ to be predicted, NAT has a collection of representations $\Omega_{u,v}^t = \left\{ \text{DE}_{uv}^t(a) \oplus Q_{uv}^t(a) | a \in \mathcal{N}_{u,v}^t \right\}$.

Ultimately, we propose to use attention to aggregate the collected representations in $\Omega_{u,v}^t$ to make the final prediction for the link $(u, v, t)$. Let MLP denote a multi-layer perceptron and we adopt

$$\text{logit} = \text{MLP}\left( \sum_{h \in \Omega_{u,v}^t} \alpha_h \text{MLP}(h) \right), \text{ where } \{\alpha_h\} = \text{softmax}(\{w^T \text{MLP}(h) | h \in \Omega_{u,v}^t\}), \quad (3)$$

where $w$ is a learnable vector parameter and the logit can be plugged in the cross-entropy loss for training or compared with a threshold to make the final prediction.

## 5  Experiments

In this section, we evaluate the performance and the scalability of NAT against a variety of baselines on real-world temporal networks. We further conduct ablation study on relevant modules and

| Measurement | Wikipedia | Reddit | Social E. 1 m. | Social E. | Enron | UCI | Ubuntu | Wiki-talk |
|---|---|---|---|---|---|---|---|---|
| nodes | 9,227 | 10,985 | 71 | 74 | 184 | 1,899 | 159,316 | 1,140,149 |
| temporal links | 157,474 | 672,447 | 176,090 | 2,099,519 | 125,235 | 59,835 | 964,437 | 7,833,140 |
| static links | 18,257 | 78,516 | 2,457 | 4486 | 3,125 | 20,296 | 596,933 | 3,309,592 |
| node & link attributes | 172 & 172 | 172 & 172 | 0 & 0 | 0 & 0 | 0 & 0 | 0 & 0 | 0 & 0 | 0 & 0 |
| bipartite | true | true | false | false | false | true | false | false |

**Table 1:** Summary of dataset statistics.

| Task | Method | Wikipedia | Reddit | Social E. 1 m. | Social E. | Enron | UCI | Ubuntu | Wiki-talk |
|---|---|---|---|---|---|---|---|---|---|
| Inductive | CAWN | $98.52 \pm 0.04$ | $98.19 \pm 0.03$ | $84.42 \pm 1.89$ | $87.71 \pm 3.26$ | $93.28 \pm 0.01$ | $\mathbf{93.67 \pm 0.65}$ | $50.00 \pm 0.00$ | $80.21 \pm 7.49$ |
| | JODIE | $95.58 \pm 0.37$ | $95.96 \pm 0.29$ | $80.61 \pm 1.55$ | $81.13 \pm 0.52$ | $81.69 \pm 2.21$ | $86.13 \pm 0.34$ | $56.68 \pm 0.49$ | $65.89 \pm 4.72$ |
| | DyRep | $94.72 \pm 0.14$ | $97.04 \pm 0.29$ | $81.54 \pm 1.81$ | $52.68 \pm 0.11$ | $77.44 \pm 2.28$ | $68.38 \pm 1.30$ | $53.25 \pm 0.03$ | $51.87 \pm 0.93$ |
| | TGN | $98.01 \pm 0.06$ | $97.76 \pm 0.05$ | $\underline{86.00 \pm 0.70}$ | $67.01 \pm 10.3$ | $75.72 \pm 2.55$ | $83.21 \pm 1.16$ | $62.14 \pm 3.17$ | $56.73 \pm 2.88$ |
| | TGN-pg | $94.91 \pm 0.35$ | $94.34 \pm 3.22$ | $63.44 \pm 3.54$ | $\underline{88.10 \pm 4.81}$ | $69.55 \pm 1.62$ | $86.36 \pm 3.60$ | $\underline{79.44 \pm 0.85}$ | $\underline{85.35 \pm 2.96}$ |
| | TGAT | $97.25 \pm 0.18$ | $96.69 \pm 0.11$ | $54.66 \pm 0.66$ | $50.00 \pm 0.00$ | $57.09 \pm 0.89$ | $70.47 \pm 0.59$ | $54.73 \pm 4..94$ | $71.04 \pm 3.59$ |
| | **NAT** | $\mathbf{98.55 \pm 0.09}$ | $\mathbf{98.56 \pm 0.21}$ | $\mathbf{91.82 \pm 1.91}$ | $\mathbf{95.16 \pm 0.66}$ | $\mathbf{94.94 \pm 1.15}$ | $\underline{92.58 \pm 1.86}$ | $\mathbf{90.35 \pm 0.20}$ | $\mathbf{93.81 \pm 1.16}$ |
| Transductive | CAWN | $98.62 \pm 0.05$ | $98.66 \pm 0.09$ | $85.42 \pm 0.19$ | $92.81 \pm 0.58$ | $91.46 \pm 0.35$ | $\underline{94.18 \pm 0.16}$ | $50.00 \pm 0.00$ | $85.50 \pm 9.70$ |
| | JODIE | $96.15 \pm 0.36$ | $97.29 \pm 0.05$ | $77.02 \pm 1.11$ | $69.30 \pm 0.21$ | $83.42 \pm 2.63$ | $91.09 \pm 0.69$ | $60.29 \pm 2.66$ | $75.00 \pm 4.90$ |
| | DyRep | $95.81 \pm 0.15$ | $98.00 \pm 0.19$ | $76.96 \pm 4.05$ | $51.14 \pm 0.24$ | $78.04 \pm 2.08$ | $72.25 \pm 1.81$ | $52.22 \pm 0.02$ | $62.07 \pm 0.06$ |
| | TGN | $\underline{98.57 \pm 0.05}$ | $98.70 \pm 0.03$ | $\underline{88.72 \pm 0.65}$ | $69.39 \pm 10.50$ | $80.87 \pm 4.37$ | $89.53 \pm 1.49$ | $53.80 \pm 2.23$ | $66.01 \pm 4.79$ |
| | TGN-pg | $97.26 \pm 0.10$ | $\underline{98.62 \pm 0.07}$ | $66.39 \pm 6.90$ | $64.03 \pm 8.97$ | $80.85 \pm 2.70$ | $91.47 \pm 0.29$ | $\underline{90.56 \pm 0.44}$ | $\underline{94.16 \pm 0.09}$ |
| | TGAT | $96.65 \pm 0.06$ | $98.19 \pm 0.08$ | $58.10 \pm 0.47$ | $50.00 \pm 0.00$ | $61.25 \pm 0.99$ | $77.88 \pm 0.31$ | $55.46 \pm 5.47$ | $78.43 \pm 2.15$ |
| | **NAT** | $\mathbf{98.68 \pm 0.04}$ | $\mathbf{99.10 \pm 0.09}$ | $\mathbf{90.20 \pm 0.20}$ | $\mathbf{94.43 \pm 1.67}$ | $\mathbf{92.42 \pm 0.09}$ | $\mathbf{94.37 \pm 0.21}$ | $\mathbf{93.50 \pm 0.34}$ | $\mathbf{95.82 \pm 0.31}$ |

**Table 2:** Performance in average precision (AP) (mean in percentage $\pm$ 95% confidence level). **Bold font** and underline highlight the best performance and the second best performance on average.

hyperparameter analysis. Unless specified for comparison, the hyperparameters of NAT (such as $M_1, M_2, F, \alpha$) are detailed in Appendix C and Table 7 (in the Appendix).

## 5.1 Experimental setup

**Datasets.** We use seven real-world datasets that are available to the public, whose statistics are listed in Table 1. Further details of these datasets can be found in Appendix B. We preprocess all datasets by following previous literatures. We transform the node and edge features of Wikipedia and Reddit to 172-dim feature vectors. For other datasets, those features will be zeros since they are non-attributed. We split the datasets into training, validation and testing data according to the ratio of 70/15/15. For inductive test, we sample the unique nodes in validation and testing data with probability 0.1 and remove them and their associated edges from the networks during the model training. We detail the procedure of inductive evaluation for NAT in Appendix C.1.

**Baselines.** We run experiments against 6 strong baselines that give the SOTA approaches for modeling temporal networks. Out of the 6 baselines, CAWN [34], TGAT [29] and TGN [20] need to sample neighbors from the historical events, while JODIE [28], DyRep [27], keep track of dynamic node representations to avoid sampling. CAWN is the only model that constructs neighborhood structural features. As we are interested in both prediction performance and model scalability, we include an efficient implementation of TGN sourced from Pytorch Geometric (TGN-pg), a library built upon PyTorch including different variants of GNNs [57]. TGN is slower than TGN-pg because TGN in [20] does not process a batch fully in parallel while TGN-pg does. Additional details about the baselines can be found in appendix C. Finally, we note that there is one concurrent work named TGL [47], and we study it in appendix E.

Regarding hyperparameters, if a dataset has been tested by a baseline, we use the set of hyperparameters that are provided in the corresponding paper. Otherwise, we tune the parameters such that similar components have sizes in the same scale. For example, matching the number of neighbors sampled and the embedding sizes. We also fix the training and inference batch sizes so that the comparison of training and inference time can be fair between different models. For training, since CAWN uses 32 as the default while others use 200, we decide on using 100 that is between the two. For validation and testing, we use batch size 32 over all baselines. We also apply the early stopping strategy for all models to record the number of epochs to converge and the total model running time to converge. We also set a time limit of 10 hours for training. once that time is reached, we will use the best epoch so far for evaluation. More detailed hyperparameters are provided in Appendix C.

**Hardware.** We run all experiments using the same device that is equipped with eight Intel Core i7-4770HQ CPU @ 2.20GHz with 15.5 GiB RAM and one GPU (GeForce GTX 1080 Ti).

| | Method | Train | Test | Total | RAM | GPU | Epoch | | Method | Train | Test | Total | RAM | GPU | Epoch |
|---|---|---|---|---|---|---|---|---|---|---|---|---|---|---|---|
| **Wikipedia** | CAWN | 1,006 | 174 | 11,845 | 30.2 | 58.0 | 6.7 | **Ubuntu** | CAWN | 1,066 | 222 | 5,385 | 38.9 | 17.4 | 1.0 |
| | JODIE | 28.8 | 30.6 | 1,482 | 28.3 | 17.9 | 19.1 | | JODIE | 6,670 | 2,860 | 76,220 | 35.3 | 18.7 | 5.5 |
| | DyRep | 32.4 | 32.5 | 1,681 | 28.3 | 17.8 | 21.5 | | DyRep | 2,195 | 2,857 | 39,148 | 38.5 | 16.6 | 1.0 |
| | TGN | 37.1 | 33.0 | 2,047 | 28.3 | 19.3 | 23.1 | | TGN | 5,975 | 2,391 | 73,633 | 39 | 19.6 | 5.5 |
| | TGN-pg | 24.2 | 6.04 | 624.8 | 30.8 | 18.1 | 15.6 | | TGN-pg | 188.7 | 36.5 | 3,682 | 37.0 | 32.1 | 11.4 |
| | TGAT | 225 | 63.0 | 3,657 | 28.5 | 24.6 | 12.0 | | TGAT | 887 | 330 | 18,431 | 47.3 | 17.0 | 2.5 |
| | **NAT** | **21.0** | 6.94 | **154.4** | 29.1 | 12.1 | 2.6 | | **NAT** | **125.8** | 41.2 | **1,321** | 28.9 | 10.1 | 5.4 |
| **Reddit** | CAWN | 2,983 | 812 | 17,056 | 38.8 | 41.2 | 16.3 | **Wiki-talk** | CAWN | 13,685 | 2,419 | 34,368 | 99.1 | 19.4 | 1.0 |
| | JODIE | 234.4 | 176 | 8,082 | 36.4 | 23.7 | 15.3 | | JODIE | 284,789 | 145,909 | 566,607 | 58.2 | 20.9 | 1.0 |
| | DyRep | 252.9 | 184 | 7,716 | 33.3 | 24.3 | 12.7 | | DyRep | 280,659 | 135,491 | 514,621 | 84.4 | 49.6 | 1.0 |
| | TGN | 271.7 | 189 | 8,487 | 33.7 | 25.4 | 15.3 | | TGN | 281,267 | 136,780 | 534,827 | 77.9 | 24.1 | 1.0 |
| | TGN-pg | 155.1 | 27.1 | 2,142 | 39.2 | 23.6 | 6.6 | | TGN-pg | 1,236 | 311.5 | 12,761 | 60.9 | 59.0 | 5.1 |
| | TGAT | 1,203 | 291 | 16,462 | 37.2 | 31.0 | 8.4 | | TGAT | 6,164 | 2,451 | 186,513 | 65.0 | 17.6 | 16.0 |
| | **NAT** | **90.6** | 28.5 | **771.3** | 37.7 | 18.5 | 3.0 | | **NAT** | **833.1** | 280.1 | **7,802** | 37.1 | 22.3 | 2.7 |

**Table 3:** Scalability evaluation on Wikipedia, Reddit, Ubuntu and Wiki-talk.

**Evaluation Metrics.** For prediction performance, we evaluation all models with Average Precision (AP) and Area Under the ROC curve (AUC). In the main text, the prediction performance in all tables is evaluated in AP. The AUC results are given in the appendix. All results are summarized based on 5 time independent experiments. For computing performance, the metrics include (a) average training and inference time (in seconds) per epoch, denoted as **Train** and **Test** respectively, (b) averaged total time (in seconds) of a model run, including training of all epochs, and testing, denoted as **Total**, (c) the averaged number of epochs for convergence, denoted as **Epoch**, (d) the maximum GPU memory and RAM occupancy percentage monitored throughout the entire processes, denoted as **GPU** and **RAM**, respectively. We ensure that there are no other applications running during our evaluations.

## 5.2   Results and Discussion

Overall, our method achieves SOTA performance on all 7 datasets. The modeling capacity of NAT exceeds all of the baselines and the time complexities of training and inference are either lower or comparable to the fastest baselines. Let us provide the detailed analysis next.

**Prediction Performance.** We give the result of AP in Table 2 and AUC in Appendix Table 6.

On Wikipedia and Reddit, a lot of baselines achieve high performance because of the valid attributes. However, NAT still gains marginal improvements. On Wikipedia, Reddit, Enron and UCI, CAWN outperforms all baselines on inductive study and most baselines on transductive. We believe the reason is that it captures neighborhood structural information via its temporal random walk sampling. However, it is not performing as well on Ubuntu. Because of the sparsity of the Ubuntu network, we suspect that CAWN's sampling method would do worse in capturing common neighbors, which might be the cause of its under-performance.

TGN and its efficient implementation TGN-pg are strong baselines without constructing structure features. On both large-scale datasets Ubuntu and Wiki-talk, TGN-pg gives impressive results on transductive learning. However, NAT still outperforms it consistently. Furthermore, TGN-pg performs poorly for inductive tasks on both datasets, while NAT gains 8-11% lift for these tasks.

On Social Evolve, NAT significantly outperforms all baselines that do not construct neighborhood structural features by at least 25% on transductive and 7% on inductive predictions. From Table 1, we can see that Social Evolve has a small number of nodes but many interactions. This highlights one of the advantages of NAT on dense temporal graphs. NAT keeps the neighborhood representation for a node's every individual neighbor separately so the older interactions are not squashed with the more recent ones into a single representation. Pairing with N-caches, NAT can effectively denoise the dense history and extract neighborhood features.

**Scalability.** Table 3 shows that NAT is always trained much faster than all baselines. The inference speed of NAT is significantly faster than CAWN that can also constructs neighborhood structural features, which achieves 25-29 times speedup on inference for attributed networks. NAT also achieves at least four times faster inference than TGN, JODIE and DyRep. Compared to TGN-pg, NAT achieves comparable inference time in most cases while achieves about 10% speed up over the largest dataset Wiki-talk. This is because when the network is large, online sampling of TGN-pg may dominate the time cost. We may expect NAT to show even better scalability for larger networks. Moreover, on the two large networks Ubuntu and Wiki-talk, NAT requires much less GPU memory. Note that albeit with just comparable or slightly better scalability, over all datasets, NAT significantly outperform TGN-pg in prediction performance.

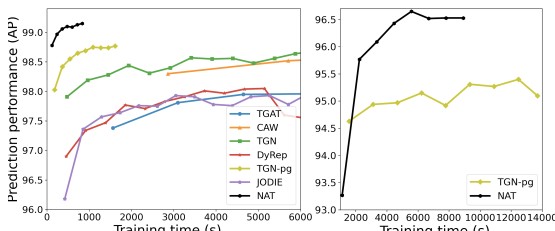
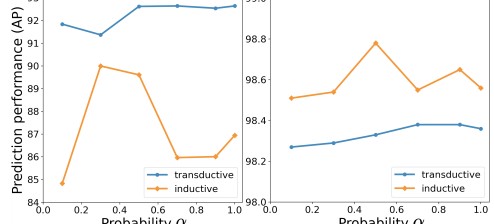

**Figure 3:** Convergence v.s. wall-clock time on Reddit (left) and Wiki-talk (right). Each dot on the curves gets collected per epoch.

**Figure 4:** Sensitivity (mean) of the overwriting probability $\alpha$ for hash-map collisions on Ubuntu (Left) & Reddit (Right).

| Ablation | Dataset | Inductive | Transductive | Train | Test | GPU |
|---|---|---|---|---|---|---|
| original method | Social E. | $95.16 \pm 0.66$ | $91.75 \pm 0.37$ | 281.0 | 89.0 | 8.88 |
| | Ubuntu | $90.35 \pm 0.20$ | $93.50 \pm 0.34$ | 125.8 | 41.2 | 10.1 |
| | Wiki-talk* | $93.81 \pm 1.16$ | $95.00 \pm 0.31$ | 833.1 | 280.1 | 22.3 |
| remove 2-hop N-cache | Social E. | $94.30 \pm 0.90$ | $90.77 \pm 0.26$ | 253.1 | 75.9 | 8.87 |
| | Ubuntu | $89.45 \pm 1.04$ | $93.48 \pm 0.34$ | 111.3 | 35.7 | 9.95 |
| remove 1-&-2-hop N-cache | Social E. | $55.10 \pm 11.54$ | $62.12 \pm 3.53$ | 212.9 | 64.0 | 8.46 |
| | Ubuntu | $85.11 \pm 0.23$ | $91.89 \pm 0.09$ | 98.1 | 29.5 | 9.07 |
| | Wiki-talk | $86.54 \pm 3.87$ | $94.89 \pm 1.83$ | 409.5 | 125.4 | 16.2 |

**Table 4:** Ablation study on N-caches. *Original method for Wiki-talk does not use the second-hop N-cache.

| Param | Size | Inductive | Transductive | Train | Test | GPU |
|---|---|---|---|---|---|---|
| $M_1$ | 4 | $92.95 \pm 2.95$ | $95.26 \pm 0.49$ | 834.9 | 281.4 | 18.4 |
| | 8 | $\mathbf{93.96 \pm 0.91}$ | $95.39 \pm 0.28$ | 806.3 | 274.9 | 19.9 |
| | 12 | $92.67 \pm 0.82$ | $95.05 \pm 0.58$ | 818.2 | 277.6 | 21.0 |
| | 16 | $93.81 \pm 1.16$ | $95.82 \pm 0.31$ | 833.1 | 280.1 | 22.3 |
| | 20 | $93.40 \pm 0.50$ | $\mathbf{95.83 \pm 0.44}$ | 841.3 | 284.8 | 23.8 |
| $M_2$ | 0 | $93.81 \pm 1.16$ | $95.82 \pm 0.31$ | 833.1 | 280.1 | 22.3 |
| | 2 | $92.91 \pm 1.01$ | $96.08 \pm 0.34$ | 960.5 | 330.9 | 22.7 |
| | 4 | $94.26 \pm 0.89$ | $\mathbf{96.29 \pm 0.09}$ | 935.3 | 322.9 | 23.8 |
| | 8 | $\mathbf{94.53 \pm 0.51}$ | $95.90 \pm 0.07$ | 943.3 | 325.3 | 26.0 |
| F | 2 | $90.86 \pm 2.52$ | $95.74 \pm 0.27$ | 843.6 | 284.0 | 18.5 |
| | 4 | $\mathbf{93.81 \pm 1.16}$ | $\mathbf{95.82 \pm 0.31}$ | 833.1 | 280.1 | 22.3 |
| | 8 | $93.55 \pm 0.93$ | $95.63 \pm 0.30$ | 828.7 | 281.1 | 26.2 |

**Table 5:** Sensitivity of N-cache sizes on Wiki-talk.

Across all datasets, NAT does not need larger model sizes than baselines to achieve better performances. More impressively, we observe that NAT uniformly requires fewer epochs to converge than all baselines, especially on larger datasets. It can be attributed to the inductive power given by the joint structural features. Because of this, the total runtime of the model is much shorter than the baselines on all datasets. Specifically, on large datasets, Ubuntu and Wiki-talk, NAT is more than three times as fast as TGN-pg. We also plot the curves on the model convergence v.s. CPU/GPU wall-clock time on Reddit and Wiki-talk for comparison in Fig. 3.

### 5.3 Further Analysis

**Ablation study.** We conduct ablation studies on the effectiveness of the N-caches. Table 4 shows the results of removing the second-hop N-caches $Z_u^{(2)}$ and removing both the first-hop and second-hop N-caches $Z_u^{(1)}, Z_u^{(2)}$. As expected, dropping the N-caches reduces the training, inference time and the GPU cost. However, it also results in prediction performance decay. Just removing $Z_u^{(2)}$ can hurt performance by up to 1%. By removing $Z_u^{(1)}$ and $Z_u^{(2)}$ but keeping only the self representation, the performance drops significantly, especially on inductive settings. Keeping only self representation is analogous to some baselines such as TGN which keeps a memory state. However, since we use a smaller dimension usually between 32 to 72, the self representation itself cannot be generalized well on these datasets. Ablation studies on other components including joint neighborhood structural features, T-encoding, RNNs, and DE are detailed in Table 8 (in the appendix).

**Sensitivity of the sizes of N-cache.** Since N-caches induce the major consumption of the GPU memory, we study how the memory size correlates with the model performance on Wiki-talk. We compare the performances between different values of $M_1$, $M_2$ and $F$ of N-caches. The baseline has $M_1 = 16$, $M_2 = 0$ and $F = 4$ and we study each parameter by fixing the other two. Table 5 details the changes in the model performance. We also study for the ubuntu dataset in Appendix Table 9.

We can see that GPU memory cost scales close to a linear function for all param changes. However, increasing the model size does not necessarily improve the performance. Changing $M_1$ to either a smaller or a larger value may decrease both the transductive and the inductive performance. Increasing $M_2$ could boost the performance, but in general, changing $M_2$ is less sensitive than changing $M_1$. Lastly, a larger $F$ could overfit the model as we can see a slight drop in the inductive prediction with the largest $F$. Overall, training and inference time remains stable because of the parallelization of NAT. Interestingly, with larger $M_1$ and $M_2$, we sometimes even see a decrease in running time. We hypothesize it is because it avoids hash collisions and short-circuits N-cache overwriting steps.

**Sensitivity of overwriting probability** $\alpha$. We also experiment on $\alpha$ to study whether N-cache refresh frequency is related to the prediction quality. Here, we use a large dataset Ubuntu and a medium dataset Reddit. Results can be found in Fig. 4. For Ubuntu, we update from the original sizes to $M_1 = 4$, $M_2 = 1$, $F = 4$ and for Reddit, we change to $M_1 = 16$, $M_2 = 2$, $F = 8$ to increase the number of potential collisions so that the effect of $\alpha$ can be better observed. On both datasets, we can see an overall trend that a larger $\alpha$ gives a better transductive performance. However, if $\alpha = 1$ and we always replace old neighbors, it is slightly worse than the optimal $\alpha$. This pattern shows that the neighborhood information has to keep updated in order to gain a better performance. Some randomness can be useful because it preserves more diverse time ranges of interactions. The inductive performance is relatively more sensitive to the selection of $\alpha$. We do not find a case when having two different probabilities for replacing $Z_u^{(1)}$ and $Z_u^{(2)}$ significantly benefits model performance, so we use a single $\alpha$ for N-caches of different hops to keep it simple.

## 6 Conclusion and Future Works

In this work, we proposed NAT, the first method that adopts dictionary-type representations for nodes to track the neighborhood of nodes in temporal networks. Such representations support efficient construction of neighborhood structural features that are crucial to predict how temporal network evolves. NAT also develops N-caches to manage these representations in a parallel way. Our extensive experiments demonstrate the effectiveness of NAT in both prediction performance and scalability. In the future, we plan to extend NAT to process even larger networks that the GPU memory cannot hold the entire networks.

## 7 Acknowledgement

Y. Luo and P. Li are partially supported by the JPMorgan Faculty Award and the National Science Foundation (NSF) award OAC-2117997. The authors also would like to thank Yanbang Wang at Cornell for the time to discuss the code for CAWN.

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

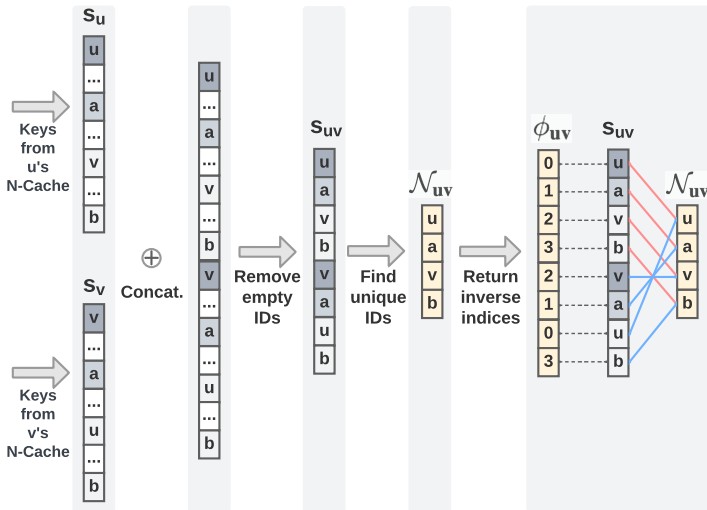

**Figure 5:** The procedure to find unique node IDs and the indices for pooling, which are used for parallel construction of DEs and joint representations.

---

**Algorithm 2:** Construct Joint Neighborhood Features ($Z_u^{(k)}, Z_v^{(k)}$ **for** $k \in \{0, 1, 2\}$)

---

1 $\text{KEY}_{uv} \leftarrow \text{concat}(s_u^{(k)}$ **for** $k \in \{0, 1, 2\}, s_v^{(k)}$ **for** $k \in \{0, 1, 2\})$;
2 $\text{VALUE}_{uv} \leftarrow \text{concat}(\text{value}(Z_u^{(k)})$ **for** $k \in \{0, 1, 2\}, \text{value}(Z_v^{(k)})$ **for** $k \in \{0, 1, 2\})$;
3 $s_{uv} \leftarrow \text{Remove EMPTY from KEY}_{uv}$;
4 Remove the corresponding EMPTY entries from $\text{VALUE}_{uv}$;
5 $\mathcal{N}_{uv} \leftarrow \text{unique}(s_{uv})$, $\phi_{uv} \leftarrow$ the index in $\mathcal{N}_{uv}$ for each of $s_{uv}$;
6 Initialize $Q_{uv}$ with $\text{length}(\mathcal{N}_{uv})$ vectors as seen in Eq (2); // to aggregate nbr. representations.
7 Scatterly add $\text{VALUE}_{uv}$ into $Q_{uv}$ according to indices $\phi_{uv}$;
8 Initialize $\text{DE}_u$, $\text{DE}_v$ with $\text{length}(\mathcal{N}_{uv})$ vectors;
9 **for** $i$ *from 0 to length($\mathcal{N}_{uv}$), in parallel (implement with scatter add using indices $\phi_{uv}$), **do**
10     **for** $w \in u, v$ **do**
11         $\text{DE}_w[i] \leftarrow [\textbf{if } \mathcal{N}_{uv}[i]$ is one of $s_w^{(k)}$ **then** 1 **else** 0 **for** $k \in \{0, 1, 2\}]$;
12 Return $\text{concat}(\text{DE}_u, \text{DE}_v, Q_{uv})$ along the last dimension;

---

## A Efficient Joint Neighborhood Features Implementation

Here, we detail the efficient implementation that generates joint neighborhood structural features based on N-Caches as introduced in Sec. 4.2. This implementation is summarized in Alg. 2.

Both DE (Eq (1)) and representation aggregation (Eq (2)) can be done for multiple nodes in parallel on GPUs using PyTorch built-in functions. Specifically, for a mini-batch of temporal links $B = \{..., (u, v, t), ...\}$, NAT first collects the union of the current neighborhoods for each end-node $s_u = \oplus_{k=1}^K s_u^{(k)}$, $s_v = \oplus_{k=1}^K s_v^{(k)}$ for all $(u, v, t) \in B$. Then, NAT follows the steps of Fig. 5: (1) Remove the empty entries in the joint neighborhood $s_u \oplus s_v$ with PyTorch function **nonzero**, denoted as $s_{uv}$. (2) Find unique nodes $\mathcal{N}_{uv}$ in the joint neighborhood $s_{uv}$. (3) Generate array $\phi_{uv}$ which stores the index in $\mathcal{N}_{uv}$ for each node in $s_{uv}$. The last two steps can be computed using PyTorch function **unique** with parameter **return_inverse** set to true. (4) Compute DE features and aggregation neighborhood features via the **scatter_add** operation with indices recorded in $\phi_{uv}$. All these operations support GPU parallel computation.

## B Dataset Description

The following are the detailed descriptions of the seven datasets we tested.

| Task | Method | Wikipedia | Reddit | Social E. 1 m. | Social E. | Enron | UCI | Ubuntu | Wiki-talk |
|---|---|---|---|---|---|---|---|---|---|
| Inductive | CAWN | $98.16 \pm 0.06$ | $97.97 \pm 0.01$ | $86.19 \pm 2.94$ | $89.32 \pm 3.29$ | $94.29 \pm 0.15$ | $\mathbf{90.89 \pm 0.48}$ | $50.00 \pm 0.00$ | $80.03 \pm 7.14$ |
| | JODIE | $95.16 \pm 0.42$ | $96.31 \pm 0.16$ | $85.16 \pm 1.24$ | $86.14 \pm 0.67$ | $82.56 \pm 1.88$ | $85.02 \pm 0.38$ | $52.41 \pm 5.80$ | $65.94 \pm 4.26$ |
| | DyRep | $93.97 \pm 0.18$ | $96.86 \pm 0.29$ | $84.38 \pm 1.69$ | $49.84 \pm 0.35$ | $76.69 \pm 2.64$ | $67.36 \pm 1.47$ | $53.22 \pm 0.03$ | $50.37 \pm 0.42$ |
| | TGN | $97.84 \pm 0.06$ | $97.63 \pm 0.09$ | $\underline{88.43 \pm 0.38}$ | $70.86 \pm 10.30$ | $75.28 \pm 1.81$ | $81.65 \pm 1.44$ | $62.98 \pm 3.36$ | $59.24 \pm 2.34$ |
| | TGN-pg | $94.96 \pm 0.33$ | $94.53 \pm 3.04$ | $63.17 \pm 4.69$ | $\underline{90.24 \pm 3.72}$ | $67.99 \pm 1.78$ | $86.02 \pm 3.34$ | $\underline{74.85 \pm 1.44}$ | $\underline{83.25 \pm 2.96}$ |
| | TGAT | $97.25 \pm 0.18$ | $96.37 \pm 0.10$ | $51.23 \pm 0.69$ | $50.0 \pm 0.00$ | $55.86 \pm 1.01$ | $70.83 \pm 0.58$ | $55.73 \pm 6.47$ | $74.50 \pm 3.71$ |
| | **NAT** | $\mathbf{98.27 \pm 0.12}$ | $\mathbf{98.56 \pm 0.21}$ | $\mathbf{92.62 \pm 1.66}$ | $\mathbf{96.13 \pm 0.46}$ | $\mathbf{95.25 \pm 1.37}$ | $90.48 \pm 1.30$ | $\mathbf{87.72 \pm 0.28}$ | $\mathbf{92.73 \pm 1.35}$ |
| Transductive | CAWN | $98.39 \pm 0.08$ | $98.64 \pm 0.04$ | $86.30 \pm 0.11$ | $90.44 \pm 0.71$ | $\underline{92.32 \pm 0.26}$ | $\underline{92.01 \pm 0.18}$ | $50.00 \pm 0.00$ | $82.83 \pm 10.30$ |
| | JODIE | $96.05 \pm 0.39$ | $97.63 \pm 0.05$ | $82.36 \pm 0.87$ | $76.87 \pm 0.32$ | $85.28 \pm 2.25$ | $91.69 \pm 0.40$ | $52.61 \pm 2.50$ | $73.32 \pm 4.37$ |
| | DyRep | $95.34 \pm 0.18$ | $97.93 \pm 0.20$ | $80.58 \pm 3.55$ | $50.05 \pm 3.64$ | $79.28 \pm 1.84$ | $72.62 \pm 2.01$ | $52.38 \pm 0.02$ | $69.89 \pm 2.67$ |
| | TGN | $98.42 \pm 0.05$ | $98.65 \pm 0.03$ | $90.37 \pm 0.40$ | $73.08 \pm 9.74$ | $82.08 \pm 4.36$ | $89.54 \pm 1.58$ | $54.13 \pm 2.52$ | $76.07 \pm 5.28$ |
| | TGN-pg | $97.06 \pm 0.39$ | $\underline{98.58 \pm 0.08}$ | $66.89 \pm 7.90$ | $66.14 \pm 10.7$ | $81.23 \pm 2.80$ | $91.16 \pm 0.30$ | $\underline{89.59 \pm 0.42}$ | $\underline{93.69 \pm 0.06}$ |
| | TGAT | $96.65 \pm 0.06$ | $98.07 \pm 0.08$ | $56.98 \pm 0.53$ | $50.00 \pm 0.00$ | $62.08 \pm 1.08$ | $79.85 \pm 0.24$ | $57.23 \pm 6.55$ | $81.82 \pm 1.87$ |
| | **NAT** | $\mathbf{98.51 \pm 0.05}$ | $\mathbf{99.01 \pm 0.11}$ | $\mathbf{91.77 \pm 0.19}$ | $\mathbf{93.63 \pm 0.36}$ | $\mathbf{93.08 \pm 0.18}$ | $\mathbf{93.16 \pm 0.31}$ | $\mathbf{92.62 \pm 0.10}$ | $\mathbf{95.33 \pm 0.26}$ |

**Table 6:** Performance in AUC (mean in percentage $\pm$ 95% confidence level.) bold font and underline highlight the best performance on average and the second best performance on average. Timeout means the time of training for one epoch is more than one hour.

| Params | Wikipedia | Reddit | Social E. 1 m. | Social E. | Enron | UCI | Ubuntu | Wiki-talk |
|---|---|---|---|---|---|---|---|---|
| $M_1$ | 32 | 32 | 40 | 40 | 32 | 32 | 16 | 16 |
| $M_2$ | 16 | 16 | 20 | 20 | 16 | 16 | 2 | 0 |
| $F$ | 4 | 4 | 2 | 2 | 2 | 4 | 4 | 4 |
| $(M_1 + M_2) * F$ | 192 | 192 | 120 | 120 | 96 | 192 | 72 | 64 |
| Self Rep. Dim. | 72 | 72 | 32 | 72 | 72 | 32 | 50 | 72 |

**Table 7:** Hyperparameters of NAT.

- Wikipedia[2] logs the edit events on wiki pages. A set of nodes represents the editors and another set represents the wiki pages. It is a bipartite graph which has timestamped links between the two sets. It has both node and edge features. The edge features are extracted from the contents of wiki pages.

- Reddit[3] is a dataset of the post events by users on subreddits. It is also an attributed bipartite graph between users and subreddits.

- Social Evolution[4] records physical proximity between students living in the dormitory overtime. The original dataset spans one year. We also split out the data over a month, termed Social Evolve 1 m., and evaluate over all baselines.

- Enron[5] is a network of email communications between employees of a corporation.

- UCI[6] is a graph recording posts to an online forum. The nodes are university students and the edges are forum messages. It is non-attributed.

- Ubuntu[7] or Ask Ubuntu, is a dataset recording the interactions on the stack exchange web site Ask Ubuntu [8]. Nodes are users and there are three different types of edges, (1) user $u$ answering user $v$'s question, (2) user $u$ commenting on user $v$'s question, and (3) user $w$ commenting on user $u$'s answer. It is a relatively large dataset with more than 100K nodes.

- Wiki-talk[9] is dataset that represents the edit events on Wikipedia user talk pages. The dataset spans approximately 5 years so it accumulates a large number of nodes and edges. This is the largest dataset with more than 1M nodes.

## C   Baselines and the experiment setup

**CAWN** [34] with source code provided **here** is a very recent work that samples temporal random walks and anonymizes node identities to achieve motif information. It backtracks historical events to extract neighboring nodes. It achieves high prediction performance but it is both time-consuming and memory-intensive. We pull the most recent commit from their repository. When measuring the CPU

---

[2] http://snap.stanford.edu/jodie/wikipedia.csv

[3] http://snap.stanford.edu/jodie/reddit.csv

[4] http://realitycommons.media.mit.edu/socialevolution.html

[5] https://www.cs.cmu.edu/~./enron/

[6] http://konect.cc/networks/opsahl-ucforum/

[7] https://snap.stanford.edu/data/sx-askubuntu.html

[8] http://askubuntu.com/

[9] https://snap.stanford.edu/data/wiki-talk-temporal.html

| No. | Ablation | Task | Social E. | Ubuntu |
|---|---|---|---|---|
| 1. | remove T-encoding | inductive | -0.74 ± 1.01 | -1.54 ± 0.10 |
| | | transductive | -1.10 ± 0.31 | -1.25 ± 0.54 |
| 2. | remove RNN | inductive | -1.18 ± 0.87 | -1.19 ± 0.86 |
| | | transductive | -1.26 ± 0.50 | -5.68 ± 4.45 |
| 3. | remove attention | inductive | -0.77 ± 1.14 | -0.28 ± 0.16 |
| | | transductive | -0.39 ± 0.43 | -0.01 ± 0.20 |
| 4. | remove DE | inductive | -3.78 ± 2.14 | -5.67 ± 2.87 |
| | | transductive | -3.43 ± 1.64 | -1.55 ± 0.16 |

**Table 8:** Ablation study with other modules of NAT (changes recorded w.r.t Table 2).

| Param | Size | Inductive | Transductive | Train | Test | GPU |
|---|---|---|---|---|---|---|
| | 8 | 89.50 ± 0.37 | **93.56 ± 0.30** | 124.4 | 41.1 | 9.85 |
| $M_1$ | 16 | **90.35 ± 0.20** | 93.50 ± 0.34 | 125.8 | 41.2 | 10.1 |
| | 24 | 88.39 ± 0.46 | 93.37 ± 0.46 | 123.5 | 41.1 | 11.0 |
| | 2 | **90.35 ± 0.20** | **93.50 ± 0.34** | 125.8 | 41.2 | 10.1 |
| $M_2$ | 4 | 89.86 ± 0.46 | 93.46 ± 0.27 | 125.7 | 41.5 | 10.2 |
| | 8 | 89.33 ± 0.40 | **93.50 ± 0.27** | 124.7 | 40.9 | 10.5 |
| | 2 | 88.82 ± 1.64 | **93.51 ± 0.17** | 124.6 | 41.3 | 9.69 |
| F | 4 | **90.35 ± 0.20** | 93.50 ± 0.34 | 125.8 | 41.2 | 10.1 |
| | 8 | 90.29 ± 0.33 | 93.42 ± 0.18 | 125.2 | 41.2 | 11.0 |

**Table 9:** Sensitivity of N-cache sizes on Ubuntu.

usage, we also notice a garbage collection bug. It causes the CPU memory consumption to keep on increasing after every batch and every epoch without any decrease. We fix the bug such that CPU memory remains constant. Our metrics in Table 3 is recorded based on our bug fix. We tune with walk length either 1 or 2. For Wikipedia, Reddit and SocialEvolve we use walk length of two, and others with only first-hop neighbors. We tune sampling sizes of the first walk between 20 and 64, and the second between 1 and 32.

**JODIE** [28] with source code provided **here** is a method that learns the embeddings of evolving trajectories based on past interactions. Its backbone is RNNs. It was proposed for bipartite networks, so we adapt the model for non-bipartite temporal networks using the TGN framework. We use a time embedding module, and a vanilla RNN as the memory update module. We use 100 dimensions for its dynamic embedding which gives around the same scale as the other models and provide a fair comparison on both performance and scalability.

**DyRep** [27] with source code provided **here** proposes a two-time scale deep temporal point process model that learns the dynamics of graphs both structurally and temporally. We use 100 gradient clips, and hidden size and embedding size both 100 for a fair comparison on both performance and scalability.

**TGN** [20] with source code provided **here** is a very recent work as well. It does not perform as well as CAWN on certain datasets but it runs much more efficiently. It keeps track of a memory state for each node and update with new interactions. We train TGN with 300 dimensions in total for all of memory module, time feature and node embedding, and we only consider sampling the first-hop neighbors because it takes much longer to train with second-hop neighbors and the performance does not have significant improvements.

**TGN-pg** with source code is provided in the PyTorch Geometric library[10] **here**. **This link** gives an example use of the library code. This is the same model design as TGN. However, it is much more efficient than TGN because it is more parallelized. Like TGN, we use 300 dimensions in total for all datasets except the largest dataset Wiki-talk. Given the limited GPU memory (11 GB), we have to tune it to 75 dimensions in total such that it can fit the GPU memory.

**TGAT** with source code provided **here** is an analogy to GAT [58] for static graph, which leverages attention mechanism on graph message passing. TGAT incorporates temporal encoding to the pipeline. Similar to CAWN, TGAT also has to sample neighbors from the history. We use 2 attention heads and and 100 hidden dimensions. We tune with either 1 or 2 graph attention layers and the samping sizes between 20 and 64.

**NAT** Since our model can provide the trade-off between performance and scalability, we tune the model with an upperbound on the GPU memory we consider acceptable. Thus, the major

---

[10]https://github.com/pyg-team/pytorch_geometric

| Method | Wikipedia | Reddit | Social E. 1 m. | Social E. | Enron | UCI | Ubuntu | Wiki-talk |
|--------|-----------|--------|----------------|-----------|-------|-----|--------|-----------|
| TGN-TGL | **99.18 ± 0.26** | **99.67 ± 0.05** | 83.51 ± 1.20 | 86.14 ± 1.45 | 70.96 ± 2.98 | 86.99 ± 2.69 | 81.15 ± 0.55 | 86.60 ± 0.32 |
| NAT-2-hop | 98.68 ± 0.04 | 99.10 ± 0.09 | **90.20 ± 0.20** | **91.75 ± 0.37** | **92.42 ± 0.09** | **93.92 ± 0.15** | **93.50 ± 0.34** | - |
| NAT-1-hop | 98.60 ± 0.04 | 98.94 ± 0.08 | 88.07 ± 0.13 | 90.77 ± 0.26 | 90.67 ± 0.13 | 93.28 ± 0.17 | 93.48 ± 0.34 | **95.82 ± 0.31** |

**Table 10:** Comparison on the transductive average precisions between TGN with TGL and NAT.

| | Method | Train | Test | Total | RAM | GPU | Epoch |
|--|--------|-------|------|-------|-----|-----|-------|
| Ubuntu | TGN-TGL | **100.5** | 38.3 | 1,506 | 40.8 | 19.0 | 7.0 |
| | NAT-2-hop | 125.8 | 41.2 | 1,321 | 28.9 | 10.1 | 5.4 |
| | NAT-1-hop | 111.3 | **35.7** | **927** | 21.9 | 9.95 | 3.0 |
| Wiki-talk | TGN-TGL | **809.7** | 310.0 | 9,157 | 43.8 | 26.5 | 3.7 |
| | NAT-1-hop | 833.1 | **280.1** | **7,802** | 37.1 | 22.3 | 2.7 |

**Table 11:** Scalability evaluation on Ubuntu and Wiki-talk between TGN with TGL and NAT.

parameters we tuned are related to the N-caches size: $M_1$, $M_2$ and $F$. During tuning, we try to keep $(M_1 + M_2) * F$ the same. We make sure that NAT's GPU consumption has to be at the same level as the baselines for all datasets. For example, for the large scale dataset Wiki-talk, the estimated upperbound for GPU is based on the consumption of other baselines as presented in Table 3. The resulting hyperparameter values are given in Table 7. We tune the attention head in the final output layer from 1 to 8 and the overwriting probability for hashing collision $\alpha$ from 0 to 1. We eventually keep $\alpha = 0.9$ as it gives the good results for all datasets. Regarding the choice of RNN, we test both GRU and LSTM, but GRU performs better and runs faster.

### C.1 Inductive evaluation of NAT

Our evaluation pipeline for inductive learning is different from others with one added process. For other sampling methods such as TGN [20] and TGAT [29], when they do inductive evaluations, the entire training and evaluation data is available to be accessed, including events that are masked for inductive test. They sample neighbors of test nodes based on their historical interactions to get neighborhood information. However, NAT does not depend on sampling. Instead NAT adopts N-caches for quick access of neighborhood information. Hence, NAT cannot build up the N-caches for the masked nodes during the training stage for inductive tasks. By the end of the training, even all historical events become accessible, NAT cannot leverage them unless they have been aggregated into the N-caches. Therefore, to ensure a fair comparison, after training, NAT processes the **full** train and validation data with all nodes unmasked, and then processes the test data. Note that in this last pass over the **full** train and validation data, we do not perform training anymore.

## D Additional Experiments

**Further Ablation study.** We further conduct ablation experiments on other components related to modeling capability, as shown in Table 8. For Ab. 1, 2, 3, and 4, we remove temporal encodings, replace RNN with a linear layer, replace the final attention layer with mean aggregation, and remove distance encoding respectively. All the ablations generate worse results. For both datasets, removing distance encoding shows significant impact as it fails to learn from joint neighborhood structures. Removing RNN generally has worse performance than removing temporal encoding. We think this is because RNN is critical in encoding temporal dependencies and is able to implicitly encode temporal information given a series of edges. Overall, we conclude that these modules are helpful to some extent for achieving a high performance.

**More on Sensitivity of N-cache sizes.** We further test the sensitivity of N-cache sizes with the Ubuntu dataset as shown in Table 9. Similar to the study on Wiki-talk, the GPU memory cost scales almost linearly while the model running time fluctuates. It also shows more evidence that a larger model size does not guarantee a better prediction performance. Similar to the study on Wiki-talk, Ubuntu only needs a tiny $F$ for the model to be successful.

## E One Concurrent Work

TGL [47] is a concurrent work of this work where it has got published very recently. TGL proposes a general framework for large-scale Temporal Graph Neural Network training. It aims to maintain the same level of prediction accuracy as baseline models while providing speedups on training and evaluation. Its major contribution is to support parallelization on multiple GPUs, which enables

training on billion-scale data. The models that this framework can support include TGN [20], JODIE [28], TGAT [29], etc. However, it neither supports the joint neighborhood features nor it is extendable to our dictionary type representations. We conduct some experiments to compare TGL with our model.

We pull the TGL framework from this **repo**. We compare NAT with TGN implemented with the framework as it is the best performing model they provided. Similar to TGN, we use embedding dimensions 100 and we follow the same setup as described in Sec. 5.1. We tune the sampling neighbor size to be around 10 to 40. If different sizes generate similar accuracy, we use the smaller size for scalability comparison. We run TGN-TGL on single GPU for a fair comparison with our model. Since TGL does not support inductive learning, we only evaluate the transductive tasks. Finally, we compare TGN-TGL with not only our baseline model, but also NAT with only the 1-hop N-cache. We document the prediction performances in Table 10 and the scalability metrics in Table 11.

Although TGN-TGL gives marginally better scores on Wikipedia and Reddit, NAT performs much better on all other datasets ($5.6-21.5\%$). Even with only 1-hop N-cache, NAT achieves $4.63-19.71\%$ better performance on non-attributed datasets. We think the reason is that given that both Wikipedia and Reddit have node and edge features, the ambiguity issue in the toy example of Fig. 1 is reduced. However, for other datasets, TGN-TGL still suffers from missing capturing the structural features in the joint neighborhood.

In terms of scalability, TGN-TGL runs faster than NAT on training for both Ubuntu and Wiki-talk, though TGN-TGL still uses a greater number of epochs and therefore longer total time. On Ubuntu, when 2-hop N-cache is involved, NAT has longer inference time than TGN-TGL. However, when only 1-hop N-cache is used, TGN-TGL takes 7% and 11% longer time compared to NAT on Ubuntu and Wiki-talk respectively. TGN-TGL performs almost all training procedures in the GPU and TGN-TGL leverages the multi-core CPU to parallelize the sampling of temporal neighbors. However, because it still has to sample neighbors, TGN-TGL is slower than NAT on large networks in testing procedures.

