# OpenReview forum: "Neighborhood-aware Scalable Temporal Network Representation Learning"
_logconference.io/LOG/2022/Conference — LoG 2022 Oral_

### Official Review · Reviewer_2WMD · 2022-10-20

**Overall Score:** 8
**Confidence:** 4

**Review:**

Overview:

In this paper, the authors described the constraints of traditional GNNs in the temporal network setting where the 1-WL test hampers expressiveness of the GNNs to distinguish certain nodes that have the same structural contexts. Specifically, the authors build on top of existing works that use distance encoding (DE) [1] and causal anonymous walks on temporal networks [2] to tackle the expressiveness limitation of traditional GNNs and also scalability issues in temporal networks. The authors propose to use N-caches to keep information about k-hop neighborhoods and update it through time to refresh the relationships among pairs of nodes for link prediction. The authors claim this method is effective and efficient through experiments on several datasets and baselines.

Pros:

1. First of all, the paper is well written with good connections between motivation, methodology, existing works, and experiments sections.
2. The motivation is conveyed clearly to readers through a good toy example to demonstrate the weakness of traditional GNN methods in the temporal network context.
3. The method is somewhat novel by including a cache which is similar to the memory concept used in other temporal graph frameworks. However, the N-cache is explicitly indexed with k-hop neighborhoods which is similar to the distance encoding concept to tackle the expressiveness issue in traditional GNNs.
4. The method is scalable through the use of down-sampling for k-hop neighborhoods and parallel GPU implementation which may be superior than other methods such as causal anonymous walks from existing work.
5. There are comprehensive experiments on the classification performance, run time analysis and ablation studies to support the authors' claim.

Cons:

1. Maybe I missed this piece of information, but I am wondering how are the node attributes of temporal networks used in model training. For these datasets: Wikipedia and Reddit, the performance gap of the proposed method and baselines is not that significant. How does the presence of node features impact your proposed method in general? Should users expect material improvements only on non-attributed temporal networks?
2. Scalability is a major selling point for this paper. It would be better if the authors could include the big O time complexities of the proposed method and baselines in addition to the empirical run time table for completeness.

Overall, this paper has some good merits in the motivating the common issues from traditional GNNs in the temporal network setting and a novel framework to tackle the raised issues in existing works. The experiment results also support the authors' claim on performance and scalability. Aside from some minor cons that the authors may be able to improve, this paper is a clear accept.

[1] https://arxiv.org/pdf/2009.00142.pdf
[2] https://arxiv.org/pdf/2101.05974.pdf

---

### Official Review · Reviewer_2zQk · 2022-10-22

**Overall Score:** 8
**Confidence:** 3

**Review:**

This paper presents a scalable approach to learning in temporal networks where each node maintains a collection of representations instead of a single feature vector. The authors show performance improvements in terms of accuracy, training and test runtimes, and memory usage.
The idea of keeping multiple representations to capture structural information is straightforward. The focus of the paper is to make this approach scalable especially in temporal networks.

Feedback and comments:
- The paper is presented well with each concept explained clearly.
- The central idea behind the approach is a simple one - save structural information which evolves with time at each node. However, the execution of the idea such that it is applicable to large networks is novel and interesting.
- The ablation study clarifies the sensitivity of the approach to different hyper parameters involved - number of hops, size of each cache, feature size, and cache refresh frequency. How does the model perform when increasing the number of hops beyond 2?
- Small sized F seems to work for datasets in the paper. I assume more complex tasks might require larger F. I would be interested in the author's comment on the trade-off in model representation vs size of F vs increasing number of hops.

---

### Official Review · Reviewer_42e5 · 2022-10-25

**Overall Score:** 6
**Confidence:** 4

**Review:**

Facing that the joint neighborhood of a set of nodes often provides crucial structural information useful for predicting whether they may interact at a certain time, this paper proposes a Neighborhood-Aware Temporal (NAT) network. In my opinion, its main contributions are: 1) construction of structural features for a joint neighborhood of multiple nodes; 2) N-cache to support parallel access on GPUs. The experimental results show the superiority of NAT. So I suggest accepting this paper after some reversions.

Strength:

·A novel strategy is proposed for temporal network representation, which may be a positive impact on the graph learning community.

·The paper structure is well and is generally easy to follow.

·The experimental results are very impressive.

Weakness:

·The work is based on the claim that "In a temporal network, the joint neighborhood of a set of nodes often provides crucial structural information useful for predicting whether they may interact at a certain time". Although there are some reference papers, but I think the authors should better clarify it clearly, maybe using an example illustration and relevant descriptions.  This will enable readers to understand the motivation of this paper easily.

·In page 4,  there is"Z(k) u,a ← Z(k−1) v,a for all a ∈ key[Z(k−1) v ]". I don't know the motivation for this. Is it too simple and causing information loss for neighbours? Can authors explain the motivation?

·"representing node v" should be "representing node a" in the table (the row of No. 2) of Page 4 ? And other typos.

·The paper proposes a novel NAT model for the temporal graph. As the mainstream studies focus on GCNs, can authors compare and summarize the internal similarities and differences between these two? I think this may provide new research ideas for the community.

I hope the authors can respond to my concerns in "weakness". Thanks.

---

### Official Review · Reviewer_8BGD · 2022-10-26

**Overall Score:** 8
**Confidence:** 3

**Review:**

## Summary:
The authors present a novel learning method for temporal networks called NAT. The core of this approach is using joint neighborhood using their data structure n-cache. NAT achieves SOTA results on the common baselines.

## Reasons for score:
The paper is well-written, achieves SOTA results, and the empirical contribution is solid. However, there are a few minor issues related to the presentation, and the method needs to be better justified theoretically.


## Pros and cons
### Pros:
- The paper solves an important problem well-known in the temporal networks community (see Fig. 1),
- Great experimental evaluation, reproducibility, and code supplements

### Cons:
- Weak analysis. It is hard to understand some of the algorithm's choices and why they are necessary.
- No comparison with the SOTA method, comparison with TGL is hidden in the appendix. IMO, it should be in the main text, and the corresponding claims should be adjusted

## Questions during the rebuttal period:
1. Please, comment on the cons.
2. Do you need large-hop representations? It would be nice to show that they are important or otherwise simplify the method.
3. The selection of alpha seems important for the method's success. Did you explore other collision resolution methods to get rid of alpha?
4. def1. Only 2 nodes can interact for a given timestamp. Are there any downsides to replace $t_1 \leq t_i \leq t_n$ in the definition?
5. Could you summarize your contributions related to n-caches? Why cannot the same be done using GPU hashtables, e.g., https://github.com/nosferalatu/SimpleGPUHashTable? Why don't you use cuckoo-hashing to address collisions? Also, I am curious about how many collisions happen in reality.
6. Comparison to https://arxiv.org/pdf/2209.15059.pdf is missing (since it is very recent, IMO, it is not required. But it would be good to leave a comment about it.)

## Some Issues:
- RNN is not defined
- Figure 2: it reads like xors, but you wanted to show concatenation. Perhaps, another symbol or write it out
- L157 I guess instead of "python dictionary," you wanted to say "hash tables" there is no need to be attached to a language here
- Table 1 is not very informative and can be moved to the appendix (and you can make table 2 larger) -- it is a bit hard to read
- L601 There is no TGL in your repo
- L614-620 does not give much info if you don't provide the predictive performance. I suggest visualizing these results in some other way, including the performance
- some grammar mistakes and misprints; please fix them

---

### Official Review · Reviewer_NySk · 2022-10-30

**Overall Score:** 8
**Confidence:** 4

**Review:**

The paper proposes Neighborhood-Aware Temporal network model (NAT) for temporal network representation learning for link prediction tasks. The authors design a dictionary-type neighborhood representation and dedicated data structure N-cache (for fast parallel computing) to record down-sampled set of the neighboring nodes to construct structural features for a joint neighborhood of multiple nodes. The final representation is a combination of structural features and aggregated representations of common neighbors. Extensive experiment results and ablation studies are provided to show the effectiveness and efficiency of the proposed model.

Strengths:
1. The idea on using joint representations of neighbors and common neighbors is new
2. The proposed data structure N-cache is novel and can accelerate the training and inference procedure
3. Experiment results are very promising
4. The paper is well written and easy to follow

Weaknesses:
1. From experiments, there are cases (Wikipedia and Reddit) that NAT is not better than TGL and has close performance with CAWN. The authors argue that for those two datasets there are both node and edge features and the ambiguity problem in Figure 1 is reduced. It is better to come up with some statistics to check the severity of the ambiguity issue to support this argument.
2. Lines 321-322 is confusing as from Table 5, increasing M2 imporves inductive case

Overall I think this is a solid work with solid implementation. The performance boost is also promising. I recommend a clear accept.

---

### Meta-Review · Area_Chair_uR2y · 2022-11-16

**Confidence:** 5
**Recommendation:** Accept for spotlight

**Meta Review:**

This paper proposes a novel and scalable temporal network representation learning, which adopts dictionary-type neighborhood representations as node representation. All reviewers agree that the proposed idea on using N-caches to store neighborhood representations and construct joint neighborhood features is novel and scalable. The experimental results show the effectiveness of the proposed method. In addition, the paper is well written and easy to follow except some minor issues. Thus, the paper is ready for acceptance.

---

### Decision · Program_Chairs · 2022-11-22

Accept (Oral)